# Physicochemical and Adsorption Characteristics of Divinylbenzene-*co*-Triethoxyvinylsilane Microspheres as Materials for the Removal of Organic Compounds

**DOI:** 10.3390/molecules26082396

**Published:** 2021-04-20

**Authors:** Alicja Bosacka, Małgorzata Zienkiewicz-Strzalka, Małgorzata Wasilewska, Anna Derylo-Marczewska, Beata Podkościelna

**Affiliations:** Institute of Chemical Sciences, Faculty of Chemistry, Maria Curie-Sklodowska University, Maria Curie-Sklodowska Sq. 3, 20-031 Lublin, Poland; malgorzata.zienkiewicz@poczta.umcs.lublin.pl (M.Z.-S.); malgorzata.wasilewska@poczta.umcs.lublin.pl (M.W.); beatapod@poczta.umcs.lublin.pl (B.P.)

**Keywords:** polymer-inorganic materials, nanostructures, microspheres, adsorption kinetics

## Abstract

In this work, organic-inorganic materials with spherical shape consisting of divinylbenzene (DVB) and triethoxyvinylsilane (TEVS) were synthesized and investigated by different complementary techniques. The obtained microspheres may be applied as sorbent systems for the purification of organic compounds from water. The hybrid microspheres combine the properties of the constituents depending on the morphologies and interfacial bonding. In this work, the influence of the molar ratio composition of crosslinked monomer (DVB) and silane coupling agent (TEVS) (DVB:TEVS molar ratios: 1:2, 1:1 and 2:1) on the morphology and quality of organic-inorganic materials have been examined. The materials were analysed using small angle X-ray scattering (SAXS) analysis, low-temperature nitrogen sorption, scanning electron microscopy (SEM) and Fourier transform infrared spectroscopy (FTIR) to provide information on their structural and surface properties. Moreover, thermal analysis was performed to characterize the thermal stability of the studied materials and the adsorbent-adsorbate interactions, while adsorption kinetic studies proved the utility of the synthesized adsorbents for water and wastewater treatment.

## 1. Introduction

The development of industry and technology has contributed to the increase in demand for materials with well-defined characteristics and combining several features of different solids in one type of structure [1,2,3]. Hybrid organic-inorganic materials are the answer to the growing needs of the material market [4,5]. One of the most important actual challenges in designing these combined systems is how to keep or increase the best properties of each component while rejecting or decreasing their limitations [6]. Organic-inorganic materials based on a polymer matrix are some of the most frequently obtained types of hybrid systems because of their ease of synthesis and further modification [7,8].

Generally, polymers have many desirable physical properties including tensile strength, modulus, toughness, or viscoelasticity. Nevertheless, because of their worse mechanical and thermal properties in comparison to metals or ceramics, numerous polymers have restricted use in engineering applications [9,10]. The mechanical and thermal properties of polymers can be improved by combining them in organic-inorganic systems. The connection between the polymer and the inorganic phase improves its strength and toughness. Moreover, the inorganic component provides the mechanical and thermal stability and usually leads to many other desired chemical and physical properties of the hybrid materials [11,12,13,14,15].

Hybrid organic-inorganic materials with spherical shapes (microspheres) can be prepared by different methods (polymerization, extraction, dispersion, solvent evaporation or emulsion techniques). In this work the suspension polymerization method was used to obtain polymer-silane microspheres. Generally, suspension polymerization is a reaction in which a monomers mixture with an initiator is dispersed in a continuous aqueous phase with the addition of a small amount of a suspension agent (stabilizer) [16,17,18]. The method as well as conditions of synthesis reaction have an influence on the properties of polymer-inorganic microspheres. The adsorption properties of hybrid microspheres result from their porous structures which develop during the polymerization reaction. The amounts, shapes and sizes of pores depend on pore-forming diluents applied to expand the polymer matrix, however, the major factor in developing the surface area is the crosslinking agent (the higher amount of crosslinker, the higher the surface area) [19,20,21,22].

Water is an essential compound used for many purposes, from drinking to industrial processes. Both drinking and industrial water must in many cases be treated before use due to the presence of various harmful contaminants [23,24,25]. Among the different wastewater treatment methods, adsorption is widely used because of its simplicity, good treatment efficiency, the availability of a wide variety of adsorbents, and relatively low costs. Aliphatic or aromatic organic substances including aldehydes, amines, nitrides, pesticides, pharmaceuticals and dyes are generally compounds that are difficult to decompose in water [26,27,28,29,30,31,32]. Therefore, the development of effective, inexpensive and selective sorbents is a crucial issue. Traditional sorbents may have many limitations: they can be susceptible to contamination and microbial growth, they may have poor mechanical and thermal stability, not enough efficiency and short life-time. The solution is to combine several solids in one in order to obtain materials with improved properties [25,29].

Adsorption processes of different aromatic compounds from solutions have been widely studied to find the dependencies between the sorption effectiveness and the properties of the adsorbate and adsorbent. Generally, the adsorption effectiveness should be treated as a result of the combined properties of the adsorbent, adsorbate and solvent as well as the parameters of the adsorption process. The affinity of various substances for solid materials depends on many factors: the structure and size of the adsorbate molecules, type of functional groups, solubility, interactions (mainly intra- or inter- molecular hydrogen bonds), as well as the properties of the solvent, and the surface and structure of the adsorbents. Therefore, data interpretation is a complex matter [33,34].

In this work, divinylbenzene-*co*-triethoxyvinysilane (DVB-TEVS) microspheres were synthesized by a copolymerization reaction at three different molar ratios 1:2, 1:1 and 2:1, in which the DVB is the organic phase and TEVS, an inorganic one. The divinylbenzene molecule has an aromatic ring linked with two vinyl groups structure (Figure 18b) while the TEVS used as a crosslinking agent is an inorganic compound with formula (Si(OCH_3_)_3_CH_2_) (Figure 18a). Organic-inorganic materials similar to DVB-TEVS are described in the literature, however, there is still an ongoing search for new materials showing high efficiency and selectivity towards various groups of pollutants. The proposed organic-inorganic microspheres show promising properties, especially as adsorbents for the removal of heavy ions and organic compounds from aqueous solutions and selective materials for solid-phase extraction techniques, because of their well-developed porous structures. Also, these materials are characterized by good thermal resistance [35,36,37,38,39,40,41,42].

The obtained samples were analyzed by various techniques. Small angle X-ray diffraction analysis (SAXS) provided detailed information about the structure of these microspherical complex systems. To strengthen and confirm the correctness of the SAXS analysis, these results were correlated and completed with low-temperature nitrogen sorption analysis, scanning electron microscopy (SEM) and also Fourier transform infrared spectroscopy/attenuated total reflection (FTIR/ATR) spectroscopy. In addition, a thermal analysis was conducted to assess the thermal properties of the obtained materials. Moreover, regarding the application in water remediation systems the adsorption properties of the hybrid materials were investigated by kinetic studies.

## 2. Results and Discussion

### 2.1. Materials Characteristics

#### 2.1.1. SEM Studies

The effect of the proportions of reacting TEVS and DVB components on the properties of the newly synthesized materials was investigated via scanning electron microscopy (SEM) and further correlated and developed by SAXS and nitrogen adsorption/desorption analysis. It was found that the copolymerization of divinylbenzene with triethoxyvinylsilane by the suspension polymerization method led to obtaining well-defined polymeric-inorganic microspheres. The SEM results presented in Figure 1 show low and high magnification images of the final microspheres.

The particles of the studied inorganic-organic materials have spherical shapes with gradually changing surface smoothness and diameters ranging from 50 µm to 200 µm. Moreover, the applied polymerization conditions yielded about 80% microspheres in the range of 100–150 µm. Depending on the molar ratio of components in the reaction mixture, differences in the morphology and porosity of the final microspheres were observed. When the ratio of organic to inorganic phases was 1:2, the spherical morphology of the material is disturbed with many deviations from the ideal spherical form, the material is heterogeneous, and several defects are observed (Figure 1a–c). The most regular shape, the highest homogeneity, and the lowest porosity were observed for the DVB:TEVS = 2:1 sample (Figure 1g–i). In this case, the surface of the spheres is the most uniform and smooth. The DVB:TEVS = 1:1 sample, shows intermediate morphological and pore properties in comparison to the materials synthesized at proportions DVB:TEVS = 2:1 and DVB:TEVS = 1:2 (Figure 1d–f).

The DVB:TEVS = 1:2 spheres are characterized by the most developed surface with the highest degree of roughness. The observed differences in the surface layers of the polymer spheres depending on the fraction of individual components indicate a clear pore-forming nature of the inorganic TEVS component and its role in the formation of an extensive surface on the created materials. The use of a silane coupling agent with a branched crosslinking nature may be responsible for the assembly of an increased porous network. The TEVS contains three ethoxy groups that occupy an area close to the Si atoms, thus limiting the formation of a polymer network in their vicinity and creating a porous system. In this case, the TEVS monomer is responsible for the observed hierarchical roughness and surface vinyl terminations [43,44]. The DVB monomer and crosslinker with its simple unidirectional structure guide the materials into a more monolithic form [45,46].

Depending on the need to create materials for various applications (catalyst supports, adsorbents), strict control of the number of ingredients of different natures has a significant impact on the ultimate morphology and allows obtaining more or less heterogeneous and porous materialstructures.

#### 2.1.2. FTIR/ATR Analysis

FTIR/ATR spectroscopy was applied to assess the advancement in the phase bonding of the organic-inorganic microspheres. The building of dual systems is associated with the content of both polymeric as well as silane phases. Therefore, the characteristic vibrations from divinylbenzene and triethoxyvinylsilane have been observed (Figure 2). The spectra show bands at 1000–1110 cm^−1^ associated with the asymmetrical stretching vibrations of Si–O–C groups, which confirm the incorporation of TEVS into the polymeric structure of the synthesized material. This peak is not observed for divinylbenzene (Appendix A). The TEVS introduction into the polymeric phase has been found for all studied materials. However, the peak with the highest intensity is observed for the sample with the highest amount of triethoxyvinylsilane [33]. The existence of stretching bonds C–H of methyl (-CH_3_), ethyl (-CH_2_–CH_3_) ~2900 cm^−1^ and vinyl groups (-CH=CH_2_) ~3080 cm^−1^ and ~2998 cm^−1^ and aromatic ring of divinylbenzene ~3020 cm^−1^ are observed. The peak intensity of aromatic ring for DVB:TEVS materials is associated with the sample composition. For DVB:TEVS = 1:2 this peak intensity is the lowest. The stretching vibrations of C=C from vinyl groups are found at 1650 cm^−1^. The ring stretches of aromatic ring are visible at 1600, 1500 and 1450 cm^−1^, respectively. The deformation vibrations of methyl and ethyl groups are observed in the range of 1470–1350 cm^−1^. Moreover, in the range of 900–650 cm^−1^ is possible to see deformation bands of C-H groups at benzene ring, therefore, the presence out of plane ring and H bending vibrations at 695 and 750 cm^−1^ is detected. Besides, the out of plane bending and twisted vibrations of vinyl groups at 908 and 991 cm^−1^ are observed [36,37,38,39,40,41,42].

#### 2.1.3. SAXS Investigation

Investigation of new materials requires the detailed analysis of structural aspects due to their importance in adsorption/desorption or immobilization processes. In this part, the microstructure of organic-inorganic materials was investigated by small-angle X-ray scattering. Table 1 includes the investigated materials and selected microstructure parameters determined from SAXS analysis.

Figure 3a shows logarithmic plots of the experimental scattering intensity I(q) as a function of the modulus of the scattering vector, q, corresponding to organic-inorganic samples with different component fractions. The scattering curve is continuous and contains no pronounced extremes or peaks. The lack of a sharp interference peak on the SAXS profile suggests the absence of the regular superstructural forms of domains [47]. The experimental scattering curve includes the sum of the scattering of various phases and their interactions [48,49]. The nature of the scattering curve is similar for all studied samples. However, some differences in the level of scattering were observed. It was found that the introduction of the greater amount of TEVS into the polymeric body raises the intensity of scattering at low-angles.

The highest level of scattering was observed for DVB:TEVS = 1:2 sample, whereas the lowest was noticed for DVB:TEVS = 2:1. Due to the presence of scattering objects (polymer domains) in the materials’ body, their size and relative quantity may be responsible for the scattering capacity and observed differences. Although all tested materials have a similar type of morphology, polymer domains in the DVB:TEVS = 1:2 sample should be marked as more visible. The phenomenon can be associated with the enhanced porosity of material as a result of greater amount of TEVS. In the case of this sample (DVB:TEVS = 1:2), increased porosity is associated with a higher amount of crosslinking agent and may cause the presence of additional nanometer forms (pores) able to generate the scattering effect [50,51]. In this way, a comparison of the level of scattering by hybrid samples can provide information about the degree of surface homogeneity [52,53].

The scattering properties of organic-inorganic materials can be discussed by Porod law. In this case, the scattering effect at large q values is applied. According to Porod law, the scattering curves of particle systems and porous materials with smooth surfaces decay proportionally to q^−3^ at large angles (I(q) ≈ k_3_/q^3^ for smeared data). When Porod’s law is observed for experimental data (like in Figure 3b), it is a good indication of well-defined interfaces between regions of different electron density. Here, the scattered intensity is proportional to the reciprocal of suitable power of scattering vector (I(q)~k^4^ for non-smeared SAXS data and I(q)~k^3^ for smeared SAXS data). Figure 3b shows the calculated Porod plots for investigated samples. Porod curves display a plateau and the SAXS data asymptotically approaches a constant value. The procedure allows evaluating the Porod constant k based on the asymptotic decay of the scattering curve at the higher angles. The amplitude of the asymptote is proportional to the particle’s surface-area-to-volume ratio and concentration. The most important feature of the Porod law suggests that the Porod constant is proportional to the surface area and the square of the electron density contrast. In this case, a simple comparison of the Porod constant allows illustrating the differences in the specific surface of materials in a way that is much simpler and faster than conventional methods of testing porosity. Comparing the absolute S/V values calculated from the Porod approximation, a clear decrease in the value for the sample containing the majority of the organic phase is visible. The S/V values equal 0.061 Å^−1^, 0.048 Å^−1^ and 0.039 Å^−1^ for DVB:TEVS = 1:2, DVB:TEVS = 1:1, and DVB:TEVS = 2:1 samples, respectively. This suggests creating an extended three-dimensional surface of material containing mostly an inorganic blowing agent and a significant reduction of this surface by increasing the degree of smoothness using larger amounts of the organic phase. The estimated values of the interface surface from SAXS data (S_SAXS_) are given in Table 1 and are equal to 521, 402 and 311 m^2^/g, respectively. Comparison of the specific surface area determined by using two different methods (low-temperature nitrogen sorption method: Brunauer–Emmett–Teller (BET) and SAXS indicates a correlation between both techniques for evaluating structural characteristics generated by polymer-domains and pores. Slight differences indicate a low amount of closed porosity of the microspheres.

Small-angle X-ray scattering allows conducting statistical analysis of scatters nanostructures. Figure 4a shows the volume-weighted particle (or pore) size distribution Dv(R) from the scattering curve of an ensemble of spherical particles (or pores) with homogeneous inner electron density distribution. Here, the size of the scattering objects can be determined. In this case, all nanometer objects are very similar (they have similar sizes). The difference concerns the relative number of these objects, which is reflected in the intensity of the maxima of Dv(R) curves. For the DVB:TEVS = 1:2 sample (Figure 4a) the size of most inhomogeneities is estimated to 40 nm, however, a significant amount of these objects is in the range from 20 nm to 100 nm. The size of scattering objects was significantly lower for DVB:TEVS = 1:1 and DVB:TEVS = 2:1 samples. Here the Dv(R) function points to the significant number of objects with dimensions of 35 nm and 20–30 nm for DVB:TEVS = 1:1 and DVB:TEVS = 2:1, respectively. Moreover the radius of gyration as the mean square distance from the center of their distribution provides a measure of the overall size of the scattering objects. Here, the DVB:TEVS = 1:2 sample exhibit some cylinder geometry as cross-sectional dimension of the objects in comparison to typical spherical scatters. For DVB:TEVS = 1:1 and DVB:TEVS = 2:1 samples the fit of the PDDF function (pair distance distribution function) was satisfactory only for spherical systems (additional calculations were performed but not presented here). The roughness of the surface was evaluated by investigation of the Porod exponent (Figure 4c). The slope of a log-log plot of intensity vs. q vector shows the fractal dimension of the scattering object. At high q values, the q^−3^ function illustrates the smooth interfaces. The log-log plots of investigated samples shows the linear range of Porod range. The Porod exponent for scattering from materials is between 2 and 3 for a surface fractal in three-dimensional space. The obtained results suggest the smooth surface for DVB:TEVS = 2:1 microspheres.

#### 2.1.4. Nitrogen Low-Temperature Sorption Analysis

The porosity characteristics of polymeric-inorganic microspheres were determined on the basis of low-temperature nitrogen adsorption-desorption measurements. In Figure 5 and Figure 6 the isotherms and pore size distributions are presented for all studied DVB-TEVS materials. The structure parameters are presented in Table 2. The differences in the isotherm course and nitrogen uptake reflect an essential variation of the porous structure. The shape of adsorption isotherms indicates a small content of micropores and significant content of mesopores. The difference between the adsorption and desorption is observed for all samples as elongated hysteresis loop of H1 type. The decrease of nitrogen adsorption, specific surface area and pore total volume is well correlated with DVB:TEVS ratio, more TEVS content the higher structure parameters. The highest specific surface areas (S_BET_) are for DVB:TEVS = 1:2, the medium for DVB:TEVS = 1:1 and the lowest for DVB:TEVS = 2:1 with values: 521, 402, 316 m^2^/g, respectively. It is evident that the addition of crosslinking agent TEVS develops porous structure of organic-inorganic materials. The increase of pore size as the average value with increase of DVB content is observed, however, the differences are not significant. The pore size distribution functions for all studied materials show similar tendencies as those obtained from SAXS data. Generally, the determined values are in good agreement with data from SEM microscopy and SAXS analysis.

### 2.2. Adsorption Studies

In order to investigate the adsorption effectiveness of the synthesized microspheres the kinetic profiles for nitrobenzene (NB), 4-nitrophenol (4-NP) and phenol (P) adsorption from aqueous solutions were measured. In Figure 7, Figure 8 and Appendix A the concentration and adsorption profiles on DVB:TEVS = 2:1, DVB:TEVS = 1:2 and DVB:TEVS = 1:1 materials are compared.

Let us first discuss the differentiation of adsorbate affinity to a given material. Based on the analysis of kinetic data presented in Figure 7, one can find that for all tested materials the strongest decrease of adsorbate concentration is observed for nitrobenzene, and the lowest for phenol. This effect can be explained based on the differences in solute solubility, whereby compounds with a lower solubility in water generally have a greater affinity to the hydrophobic surface of the adsorbent [33]. Next one can discuss the differentiation of the adsorption properties of all DVB-TEVS materials towards various solutes. Basing on the analysis of data presented in Figure 8 and Appendix A, we can find the differences in the amount and rate of concentration loss from the solution of nitrobenzene, phenol and 4-nitrophenol. In the case of nitrobenzene adsorption (Figure 8a and Appendix A), the greatest adsorbate losses from the solution are noted for the experimental system with the most hydrophobic DVB:TEVS = 2:1, slightly smaller for the least hydrophobic DVB:TEVS = 1:2, and the smallest for DVB:TEVS = 1:1 with intermediate properties. In the case of 4-nitrophenol and phenol adsorption the greatest adsorbate losses are observed for DVB:TEVS = 1:1. Comparing the adsorption effectiveness of the obtained materials towards different adsorbates one can find a relatively large degree of similarity between the DVB:TEVS = 2:1 and DVB:TEVS = 1:2, however, DVB:TEVS = 1:1 shows quite different properties. Such a differentiated behaviour may be explained taking into account not only the hydrophobic/hydrophilic properties, but also the structural adsorbent characteristics and polymer swelling which can be responsible for penetration of adsorbate molecules into the material structure. As it will be discussed later basedon the thermal analysis data, the interactions of solute molecules with DVB-TEVS microspheres have physical character. Thus, the measured kinetic profiles reveal the global effect of all these factors.

The obtained experimental data were analysed using many equations and adsorption kinetics models. The relative standard deviations obtained in fitting procedure are listed in Table 3. It was observed that the best quality of fitting to the kinetic data was obtained for a multi-exponential equation, which parameters are summarized in Table 6. As can be seen, adsorption in the investigated experimental systems is a complex process, which kinetics can be described by two or three terms of the m-exp equation. The fitting quality is very good, which is confirmed by low values of relative standard deviations *SD(c)/c_o_* in the range from 0.205% to 0.805% and low values of determination factors 1-R^2^ in the range from 4.2 × 10^−4^ to 5.4 × 10^−2^. Additionally, the *u_eq_* parameter values confirm the differences in the adsorption values. Moreover, for all tested systems, the half time *t_1/2_* values are determined, which are defined as the time needed to obtain a half of the concentration change. The obtained half-time values confirm the differences in the adsorption rates of NB, P and 4-NP. Additionally, based on the analysis of the data presented in Figure 7, Figure 8 and Appendix A, and in Table 4 it may be found that the highest adsorption kinetics for all tested adsorbates was recorded in the system with the DVB:TEVS = 1:1.

Summing up the discussion on DVB-TEVS adsorption effectiveness one can find that the synthesized materials show remarkable selectivity towards various adsorbates. Depending on the synthesis procedure it is possible to obtain the materials which are characterized by divergent uptakes and kinetic characteristics.

### 2.3. Thermal Analysis

Thermal analysis is a very useful technique for the characterization of materials, which may be also helpful in investigations of adsorbate-adsorbent interactions. The thermal properties of pure polymer-silane hybrids and the materials loaded with organic adsorbates were investigated using TG, DTG and DSC techniques. In Figure 9, the TG, DTG and DSC curves measured for the synthesized DVB-TEVS materials are presented. One can state that they show similar thermal behaviour with several stages of the thermal degradation. The data obtained from TG/DTG/DSC curves are summarized in Table 5. The thermal analysis proved that up to 170 °C the materials are highly thermally stable. The exothermic peaks visible on DSC curve starting at 170 °C may be associated with the additional crosslinking processes related to the presence of tetrafunctional DVB monomer. The main step related to thermal destruction of crosslinked microspheres follows at temperatures 330–550 °C with the maximum at about 450 °C. In this stage an 84–85% decrease in the weight of studied materials is observed. In the 550–950 °C temperature range the weight loss is below 3%. The total mass loss is similar for all DVB-TEVS materials and it is about 87%. These results confirm that the materials are thermally stable up to at least 330 °C and the main destruction process occurs at about 450 °C, therefore, the synthesized materials may be applied in relatively wide temperature ranges. These observations were confirmed with the data of identification of thermal destruction products by mass spectrometry (MS) analysis. The MS profiles for DVB:TEVS = 1:1 are shown in Figure 10 and Figure 11. The MS analysis confirms that the initial decomposition of these materials happens below 200 °C, and the major material destruction process is above 330 °C. Moreover, the MS data indicate that the thermal behaviour of DVB-TEVS materials is strictly connected with the presence of the 1,4-divinylbenzene matrix. The course of TG, DTG and DSC curves as well as MS spectra for the DVB:TEVS materials (Figure 9 and Figure 10) and pure DVB (presented in Appendix A) are similar. Many signals from the defragmentation of the aromatic structure of divinylbenzene are visible: water (*m*/*z* = 18), vinyl group (*m*/*z* = 27), ethyl group (*m*/*z* = 29), benzene (*m*/*z* = 78), phenyl group (*m*/*z* = 77), benzyl group (*m*/*z* = 91). In addition, the gaseous MS profiles of thermal degradation of the studied materials confirm the presence of triethoxyvinylsilane coupling agent (Appendix A for DVB:TEVS = 1:1, Appendix A) (*m*/*z* = 189) and absence of TEVS in pure polymer.

Taking into consideration the TG, DTG and DSC curves obtained for the materials loaded with organic solutes (Figure 12, Figure 13, Figure 14 and Table 6) one can find small differences for various adsorbates. In the range of 170–330 °C the weight losses for all materials with adsorbed organic solutes are higher than in the case of pure DVB-TEVS materials (0.4–1.4%) suggesting relatively weak adsorbent-adsorbate interactions of physical nature. However, for 4-NP and P the weight loss is higher (2.6–4%) in comparison to the NB (0.7–1.3%)-containing sample.

The TG, DTG and DSC results were completed by mass spectrometry analysis. The 3D MS profiles and MS spectra for DVB:TEVS = 1:1 with adsorbed organic substances are presented in Figure 15 and Figure 16. The gaseous MS profiles of DVB-TEVS materials after adsorption (Figure 17) revealed that in the case of 4-NP certain amounts of nitrogen compounds released above 170 °C are found: nitrogen oxide (*m/z* = 30), nitrogen dioxide (*m/z* = 46) and also small amounts of nitrophenol (*m/z* = 139) and nitrobenzene (*m/z* = 123). These products of thermal decomposition confirm the adsorption 4-NP and explain the higher mass losses in the range of 170–330 °C. In the case of NB adsorption larger amounts of nitrogen compounds evaporate up to 100 °C (the MS signals recorded at low temperatures), thus, it explains lower mass losses in the range 170–330 °C in comparison to 4-NP. The main step of the destruction of microspherical structures with residues of adsorbates occurs runs at a maximum of 450 °C.

## 3. Experimental and Calculation Procedures

### 3.1. Chemicals

Triethoxyvinylsilane (TEVS), decan-1-ol and poly(vinyl alcohol) (PVA) were purchased in Fluka AG (Buchs, Switzerland). α,α′-Azoiso-bis-butyronitrile (AIBN) and DVB (62.2% of 1,4-divinylbenzene, 0.2% of 1,2-divinylbenzene and ethylvinylbenzene) were obtained from Merck (Darmstadt, Germany). Prior to use DVB and TEVS were washed with 3% aqueous sodium hydroxide solution. Acetone and toluene have been obtained from Avantor Performance Materials Poland S.A. (Gliwice, Poland). The organic substances used in kinetic experiment: phenol and 4-nitrophenol were bought from Merck and the nitrobenzene from Avantor Performance Materials Poland S.A.

The structures of TEVS and DVB, as well as the model of the hybrid chain are shown in Figure 18. The properties of organic compounds used as the adsorbates in the kinetic studies are presented in Table 7.

### 3.2. Materials Synthesis

Copolymerization of 1,4-divinylbenzene with triethoxyvinylsilane was performed in the aqueous medium by using a suspension polymerization method. In synthesis vessel 75 mL of redistilled water and 1 g of poly(vinyl alcohol) were placed and stirred for 1 h at 80 °C in a three-necked flask fitted with a mechanical stirrer, thermometer and air cooler. Then, the solutions containing DVB and a corresponding amount of TEVS, the initiator AIBN (1 wt%) and the mixture of pore-forming diluents (toluene and 1-decanol, taken in 1/1 (*v*/*v*) proportions) were added while stirring to the aqueous medium. The mixture was stirred at 350 rpm for 12 h at 85 °C. The obtained microspheres were washed with distilled water (2 L), filtered, dried and extracted in a Soxhlet apparatus with boiling acetone for 3 h. The amounts of individual components in reaction mixtures are presented in Table 8.

### 3.3. Investigation Methods

#### 3.3.1. Scanning Electron Microscopy (SEM)

The surface morphology of the samples was studied by the field emission Scanning Electron Microscopy (SEM) employing a Quanta^TM^ 3D FEG (FEI Company, Hillsboro, OR, USA) apparatus operating at 5 kV. A high vacuum (4 × 10^−4^ Pa) mode was applied for imaging the investigated samples. Prior to measurement, the samples were mounted on aluminum stubs and sputtered with gold.

#### 3.3.2. Nitrogen Adsorption-Desorption Measurements

The porosity of the materials was examined applying low-temperature nitrogen adsorption-desorption isotherms at 77 K. The values of the parameters characterizing the properties of analyzed samples were obtained: the BET specific surface area (S_BET_) (assessed from the linear BET plot of adsorption data), the total pore volume (V_t_) (from the adsorption value at the relative pressure *p/p_o_*~0.99), the micropore volume (V_mic_) (from the t-plot), and the pore size distributions (PSD) followed by the Barrett, Joyner, and Halenda (BJH) procedure and the Non-Local Density Functional Theory (NLDFT) approach (ASAP 2020 analyzer, Micromeritics, USA Before the measurement, volatile materials adsorbed on the surface were driven by thermal degassing. In this procedure, the precisely weighed amount of the samples (~0.15 g) were outgassed at 120 °C and pressure of 1 mmHg for 24 h in a degas port of analyser.

#### 3.3.3. Small-Angle X-ray Scattering (SAXS)

Small X-ray scattering (SAXS) experiments were carried out by the Empyrean X-ray diffraction platforms (PANalytical), in transmission geometry CuKα radiation with a wave-length (λ) of 1.5418 Å was used at 40 kV and 40 mA configuration at room temperature. The X-ray source consisted of an anode Cu source with a line focus type. SAXS measurements were performed using a 10 mm fixed mask, Cu 0.2 mm beam attenuator and PIXcel3D area detector. SAXS is an analytical technique that measures the X-ray intensities scattered by a sample as a function of the scattering angle (scattering vector q). Scattering vector q is given by q = 4πsin(θ/2)/λ, where λ is the wavelength of the X-ray beam and θ is the scattering angle. Measurements were made at a small angle in the range of 0.2 to 4.0deg of 2θ.

The Porod approximation was applied to SAXS data processing to assess the interface between the matrix and the distracting objects (pores). Porod constant (k_p_) can be applied for determination of the specific surface area of the two-phase system using X-ray scattering at small-angle:(1)$$\frac{S}{V}=\frac{4p\left(1-p\right)k_{p}}{Q_{p}}$$
where p and (1-p) are the volumes of two phases, k_p_ is the Porod’s constant, and Q_p_ is the Porod’s invariant which is proportional to the mean-square density fluctuation of the whole scattering volume. If S/V is calculated by D_v_(R) or Porod algorithm, the specific surface area from SAXS (S_SAXS_) can be calculated according to the equation:(2)$$S_{\mathrm{SAXS}}\left[\frac{m^{2}}{g}\right]=\frac{10000\cdot \frac{S}{V}{[Å}^{-1}]}{d\left[\frac{g}{{\mathrm{cm}}^{3}}\right]}$$
where S/V is the surface divided by volume ratio calculated from the distribution curve and d is the mass density [54,55].

#### 3.3.4. Fourier Transformed Infrared/Attenuated Total Reflection Analysis (FTIR/ATR)

Fourier Transformed Infrared connected with Attenuated Total Reflection (FTIR/ATR) analysis was conducted by using IR spectrometer TENSOR 27 (Brucker, Germany) equipped with the diamond crystal. The spectra were recorded in the spectral range of 400–4000 cm^−1^.

#### 3.3.5. Thermal Analysis

Thermal analysis was carried out on a TG/DTA/DSC apparatus (STA 449 Jupiter F1, Netzsch, Selb, Germany). Samples of mass 20 mg placed in alumina crucibles and heated from 30 to 1200 °C under helium atmosphere with flow rate 50 mL·min^−1^ and heating rate 10 °C·min^−1^. The gaseous products emitted during decomposition of materials after adsorption were analysed by Quadrupole Mass Spectroscopy (QMS 403C Aeölos, Selb, Germany). The QMS data in the range from 10 to 300 amu were recorded.

#### 3.3.6. Adsorption Studies

Kinetic measurements were conducted using a UV-Vis spectrophotometer Cary 100 (Varian, Australia) with a flow working cell for periodic measurements of a solution concentration in a closed system. In all cases, 100 mg of sample was placed in a quartz vessel connected with a stirrer (110 rpm) and thermostatic system (25 °C) and filled with 100 mL solutes. Each of the samples was treated with three aqueous solutions of organic compounds: phenol, 4-nitrophenol, and nitrobenzene. The initial concentration of solutions in all cases was equal to 0.205 mmol/L. At definite time intervals, the solution samples were gathered in a flow cell, and absorption spectra were measured, and the solution was turned back to the reaction vessel. The absorbance spectrum was recorded for wavelength ranges from 200 to 450 nm. The concentration in the function of time profiles for the experimental systems was calculated from the recorded spectra.

The kinetic curves were analysed using several equations: first-order equation (FOE), the second-order equation (SOE) and pseudo first and second-order equations (PFOE and PSOE), the so-called mixed 1,2-order equation (MOE), the fractal-like MOE equation (f-MOE), the fractal-like first-order equation (f-FOE) and the fractal-like second-order equation (f-SOE), the so-called multiexponential equation (m-exp), the intraparticle diffusion model (IDM), and the pore diffusion model (PDM). The basic and pseudo first-order equations (FOE and PFOE can be expressed by a linear relationships (3) or (4):(3)$$ln\left(a_{eq}-a\right)=ln(a_{eq}-a_{o})-k_{1}$$
where a is the temporary adsorbed amount, *a_o_* is the initial amount, *a_eq_* is the equilibrium adsorbed amount, *k_1_* is the adsorption rate coefficient.
(4)$$ln\left(c_{eq}-c\right)=ln(c_{eq}-c_{o})-k_{1}$$
where *c* is the temporary adsorbate concentration, *c_o_* is the initial concentration, *c_eq_* is the equilibrium concentration, *k_1_* is the same as in Equation (3).

The second-order and pseudo second order equations (SOE and PSOE) can be noted as:(5)$$a=a_{eq}\left[k_{2}t/\left(1+k_{2}t\right)\right]$$

The linear forms of SOE and PSOE equations are:(6)$$t/a=\left(1/a_{eq}\right)\left(1/k_{2}+t\right)$$
(7)$$a=a_{eq}-\left(1/k_{2}\right)\left(a/t\right)$$
where *k_2_ = k_2a_a_eq_* and *k_2a_* are the rate coefficients for pseudo-second order kinetics.

The generalization of the first and second order kinetics is the 1,2-mixed-order kinetic equation (MOE) which is the combination of the first and second order units and can be expressed as a relative adsorption progress in time (*F*):(8)$$F=a/a_{eq}=\frac{1-exp\left(-k_{1}t\right)}{1-f_{2}exp\left(-k_{1}t\right)}$$
(9)$$ln\left(\frac{1-F}{1-f_{2}F}\right)=-k_{1}t$$
where *f_2_* < 1 is the normalized share of the second order process in the kinetics. In some cases, the MOE equation is reduced to the simple kinetic equations of the first (*f_2_* = 0) and the second order (*f_2_* = 1) type. Non-ideality effects can be considered using fractal-like equations (f-FOE, f-SOE and f-MOE) in which fractal coefficient f is used.

The multi-exponential equation (m-exp) is applied to describe numerous first-order processes or the follow up processes and has the following forms:(10)$$c=\left(c_{o}-c_{eq}\right)\sum_{i=1}^{n}f_{i}exp\left(-k_{i}t\right)+c_{eq}$$
(11)$$c=c_{o}-c_{o}u_{eq}\sum_{i=1}^{n}f_{i}\left[1-exp\left(-k_{i}t\right)\right]$$
where *k_i_* is the rate coefficient, *u_eq_* = 1 − *c_eq_*/*c_o_* is the relative loss of adsorbate from the solution, and “*i*” is the term of m-exp equation.

The Intraparticle Diffusion Model (IDM, Crank) characterizes the adsorption processes on the spherical adsorbent grains. When the concentration of adsorbate is constant the equation can be expressed as:(12)$$F=1-\frac{6}{\pi^{2}}\overset{\infty }{\underset{n=1}{\sum }}\frac{1}{n^{2}}exp\left(\frac{-\pi^{2}\cdot n^{2}\cdot D_{a}\cdot t}{r^{2}}\right)$$
where r is the radius of adsorbent particle, *D_a_* is the effective diffusion coefficient and is given in equation:(13)$$D_{a}=\frac{D}{\tau_{p}\cdot \left(1+\rho \cdot K_{H}\cdot \varepsilon_{p}\right)}$$
where *D* is the molecular diffusion coefficient, *τ_p_* is the dimensionless pore tortuosity factor, *ρ* is the particle density, *ε_p_* is the particle porosity and *K_H_* is the Henry adsorption constant.

The Pore Diffusion Model (PDM, McKay) describe adsorption on porous solids which combines the transfer resistance of adsorbate particles through the surface layer, the proportional penetration of the adsorbate into the adsorbent grains, the sharp boundary between the space in which the equilibrium state is determined and the space without the adsorbate, and can be expressed by the mathematic formula:(14)$$\frac{dF}{d\tau_{s}}=\frac{3\left(1-u_{eq}\cdot F\right)\cdot {\left(1-F\right)}^{\frac{1}{3}}}{1-B\cdot {\left(1-F\right)}^{\frac{1}{3}}}$$
where *u_eq_* is the relative adsorbate loss, the parameter *B* = 1 − 1/*B_i_* and *B_i_* = *K_f_*/*D_p_* is the Biot number, *D_p_* is the pore diffusion coefficient, *K_f_* is the external mass transfer coefficient, *τ_s_* is the undersized model time [32,56,57].

## 4. Conclusions

Divinylbenzene-*co*-triethoxyvinysilane microspheres were synthesized at different component molar ratios (DVB:TEVS = 1:2, DVB:TEVS = 1:1 and DVB:TEVS = 2:1). The obtained materials were characterized by differentiated morphology and porosity. The DVB:TEVS = 1:2 microspheres showed many deviations from the ideal spherical form, while the DVB:TEVS = 2:1 microspheres were characterized by the most regular shape, uniform and smooth surface. FTIR/ATR spectroscopy confirmed the incorporation of TEVS into the polymeric structure of the synthesized materials. The SAXS and nitrogen adsorption-desorption techniques produced similar results regarding the structural characteristics of the copolymers. The specific surface areas determined by both methods were similar with slight differences indicating a low amount of closed porosity. The correlation between the structure parameters (specific surface area and pore total volume) and DVB:TEVS ratio was found.

The adsorption properties of the synthesized microspheres towards nitrobenzene, 4-nitrophenol and phenol were studied by kinetic measurements. Generally, the differences in the amount and rate of concentration loss from the solutions of all solutes were observed. For all tested materials the strongest decrease of adsorbate concentration was found for nitrobenzene, and the lowest for phenol which was connected with adsorbate solubility/hydrophobicity. For nitrobenzene adsorption the greatest adsorbate losses from the solution were observed in the case of the most hydrophobic DVB:TEVS = 2:1, however, for 4-nitrophenol and phenol adsorption the greatest adsorbate losses were found for DVB:TEVS = 1:1. The adsorption effectiveness of DVB:TEVS = 2:1 and DVB:TEVS = 1:2 materials towards different adsorbates was similar, however, DVB:TEVS = 1:1 showed quite different properties. The measured kinetic profiles revealed the global effect of several factors: the hydrophobic/hydrophilic properties, the structural adsorbent characteristics, the polymer swelling, the adsorbent-adsorbate interactions.

The thermal analysis proved that the synthesized DVB-TEVS materials were thermally stable up to at least 330 °C, thus, they may be applied in relatively wide temperature ranges. The weight losses for all materials with adsorbed organic solutes were higher in comparison to pure DVB-TEVS materials suggesting relatively weak adsorbent-adsorbate interactions of physical nature. The MS gaseous profiles revealed certain amounts of nitrogen compounds confirming adsorption of 4-nitrophenol and nitrobenzene.

The synthesized polymeric materials showed remarkable selectivity towards various organic adsorbates confirmed by divergent uptakes and kinetic characteristics. They seem to be promising materials for adsorption and separation techniques.

## Figures and Tables

**Figure 1 molecules-26-02396-f001:**
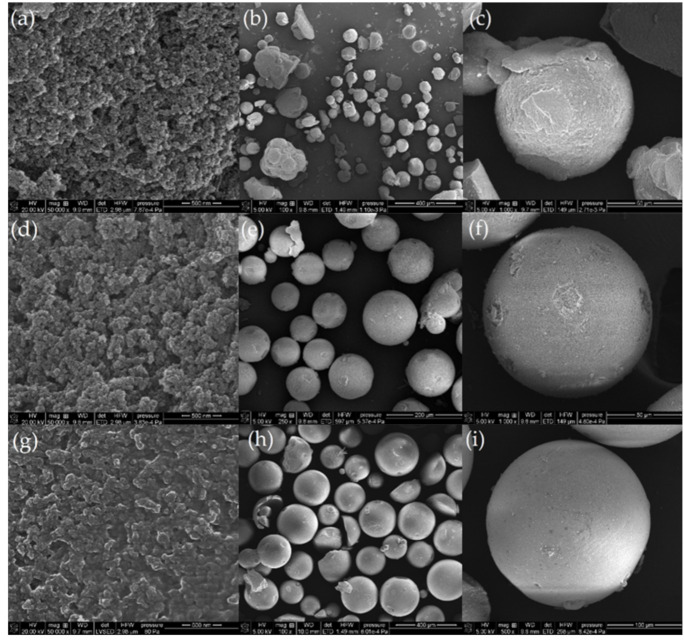
SEM images of DVB:TEVS = 1:2 (**a**–**c**), DVB:TEVS = 1:1 microspheres (**d**–**f**) and DVB:TEVS = 2:1 microspheres (**g**–**i**) in different magnification.

**Figure 2 molecules-26-02396-f002:**
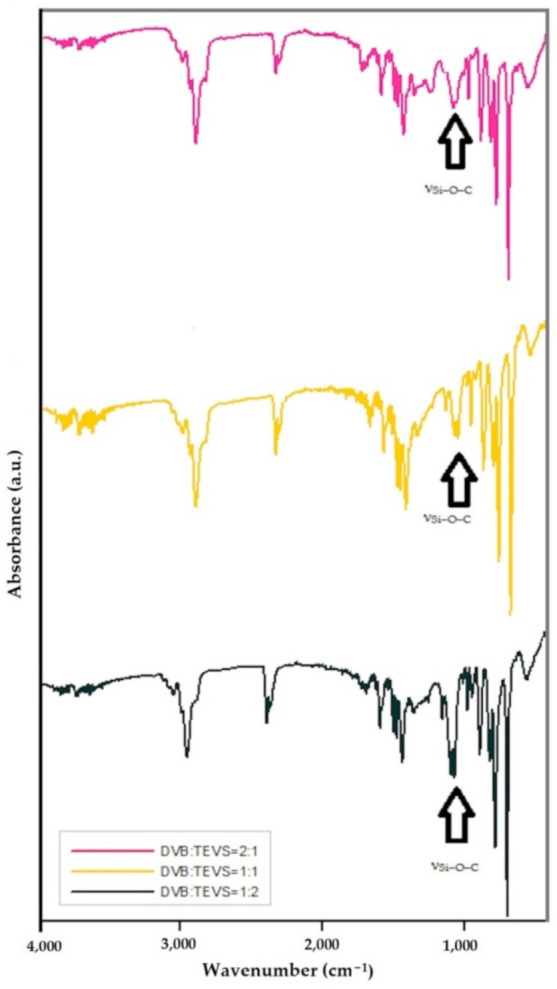
FTIR/ATR spectra for DVB:TEVS materials with phase ratios: 2:1, 1:1 and 1:2.

**Figure 3 molecules-26-02396-f003:**
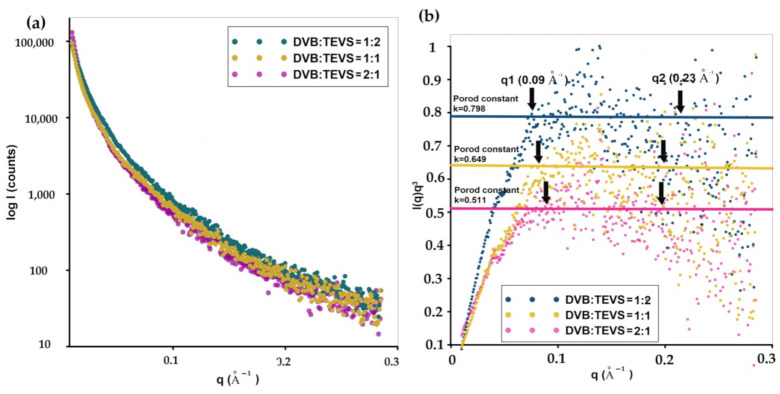
Experimental SAXS profiles corresponding to hybrid microspheres with various amounts of organic and inorganic components (**a**) Porod plots, and Porod constants determined from experimental patterns (**b**). The black arrows show the range of scattering vector suitable for the Porod linear range. The log-log plots of SAXS intensity in the power-law range (**c**).

**Figure 4 molecules-26-02396-f004:**
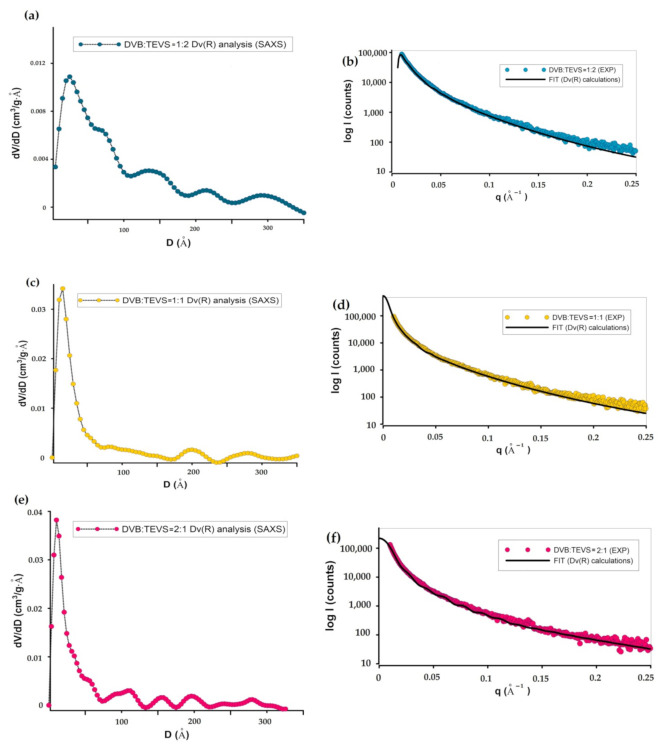
Particle Size Distribution by volume analysis Dv(R) for investigation systems: DVB:TEVS = 1:2 (**a**), DVB:TEVS = 1:1 (**c**) and DVB:TEVS = 2:1 (**e**). The insets of plots correspond to fit curves extrapolated for experimental SAXS data (**b**,**d**,**f**).

**Figure 5 molecules-26-02396-f005:**
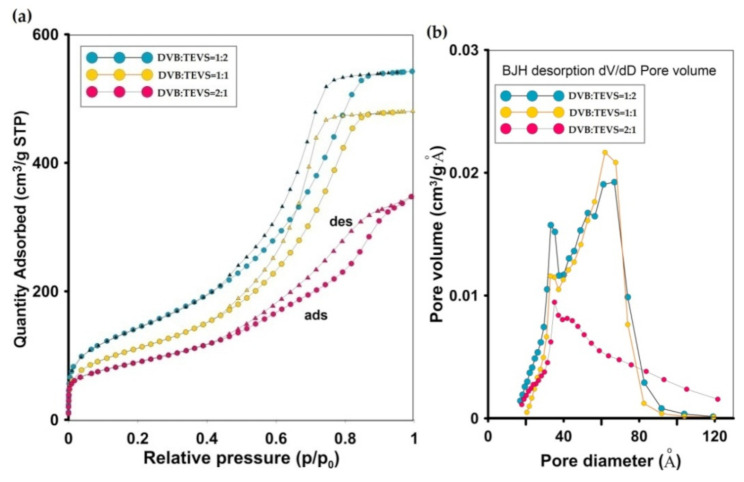
The nitrogen adsorption-desorption isotherms for DVB-TEVS materials (**a**), pore size distributions by the Barrett–Joyner–Halenda (BJH) with Halsey-Faas correction for desorption (**b**) and adsorption data (**c**).

**Figure 6 molecules-26-02396-f006:**
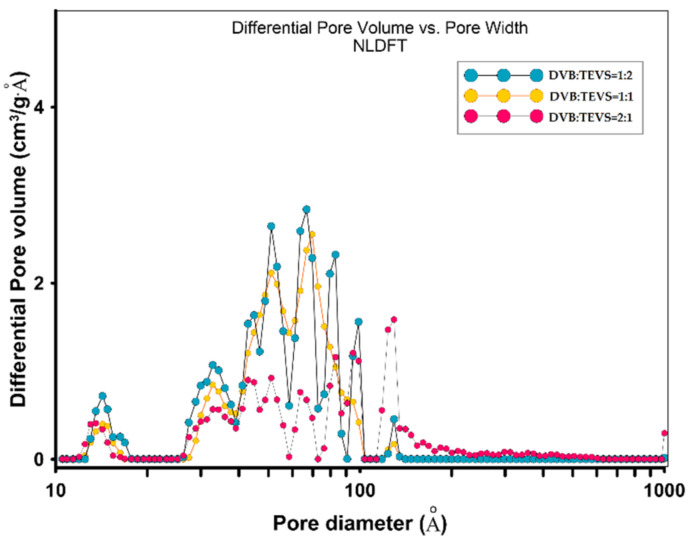
Pore size distributions obtained by Density Functional Theory with Model: N2 @ 77 K, Slit Pores. Method: Non-negative Regularization: 0.01000 and Standard Deviation of Fit: 6.62419 cm^3^/g STP.

**Figure 7 molecules-26-02396-f007:**
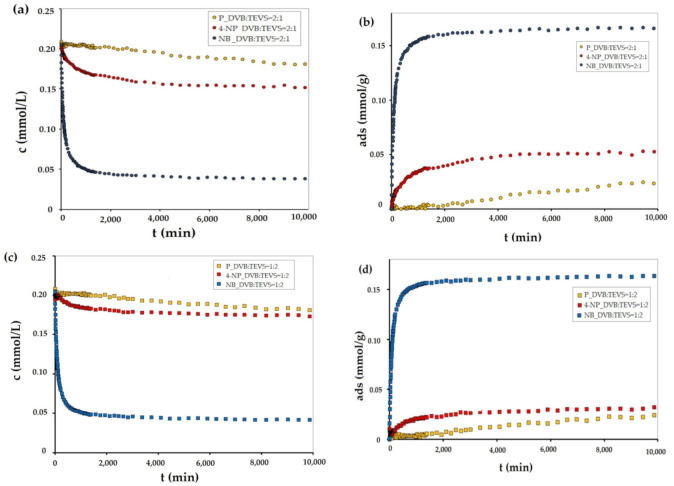
Adsorption kinetics for NB, P and 4-NP on DVB:TEVS = 2:1 (**a**,**b**), DVB:TEVS = 1:2 (**c**,**d**) and DVB:TEVS:1:1 (**e**,**f**) microspheres presented as changes in concentration over time and changes in adsorption over time.

**Figure 8 molecules-26-02396-f008:**
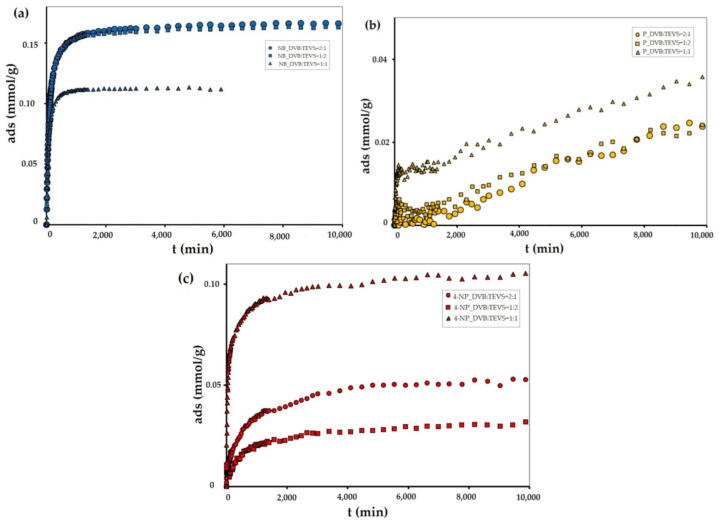
Adsorption kinetics for NB (**a**), P (**b**) and 4-NP (**c**) on DVB:TEVS = 2:1, DVB:TEVS = 1:2 and DVB:TEVS = 1:1 materials presented as changes in adsorption over time.

**Figure 9 molecules-26-02396-f009:**
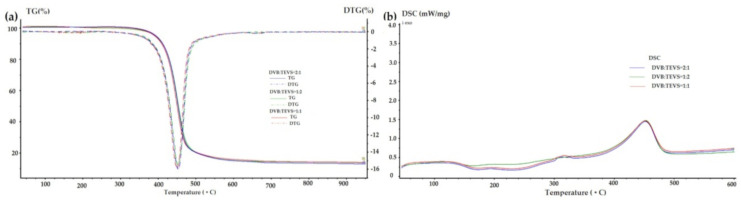
TG, DTG (**a**) and DSC (**b**) curves for DVB-TEVS materials measured in helium conditions.

**Figure 10 molecules-26-02396-f010:**
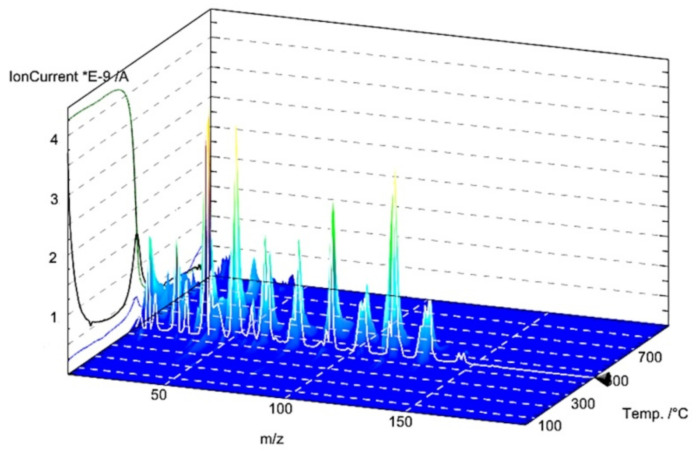
3D MS profile registered during decomposition of pure DVB:TEVS = 1:1.

**Figure 11 molecules-26-02396-f011:**
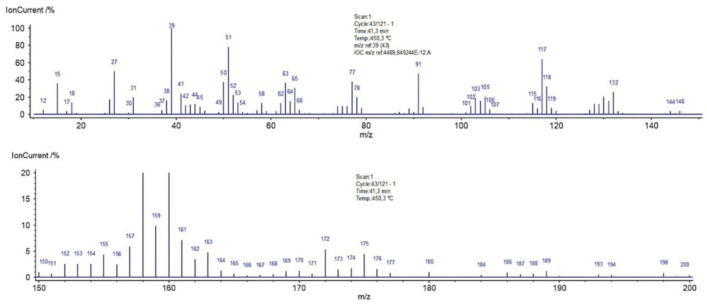
MS spectra of DVB:TEVS = 1:1 decomposition at 450 °C.

**Figure 12 molecules-26-02396-f012:**
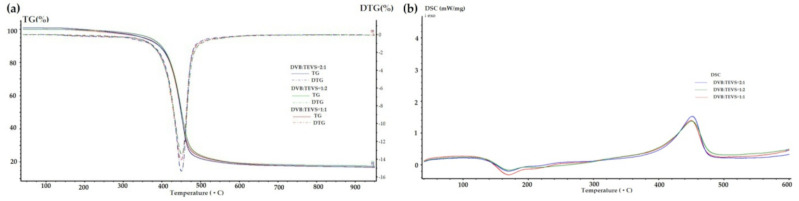
TG, DTG (**a**) and DSC (**b**) curves for DVB-TEVS materials measured in helium conditions after adsorption of P.

**Figure 13 molecules-26-02396-f013:**
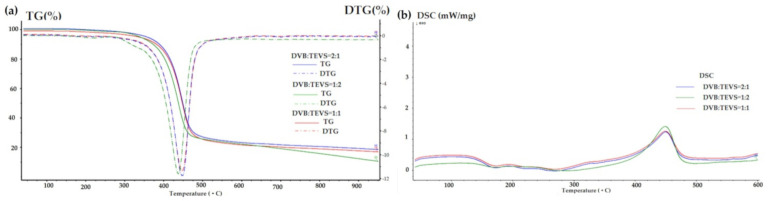
TG, DTG (**a**) and DSC (**b**) curves for DVB-TEVS materials measured in helium conditions after adsorption of 4-NP.

**Figure 14 molecules-26-02396-f014:**
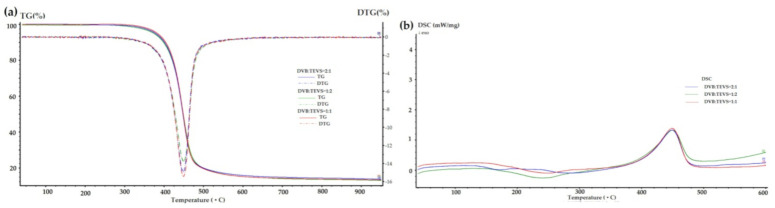
TG, DTG (**a**) and DSC (**b**) curves for DVB-TEVS materials measured in helium conditions after adsorption of NB.

**Figure 15 molecules-26-02396-f015:**
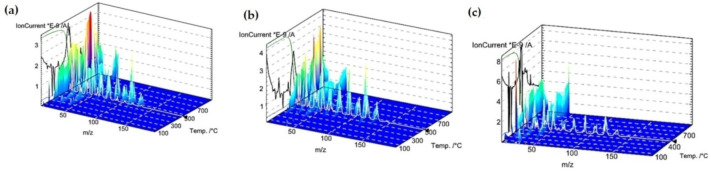
3D MS profiles registered during decomposition of DVB:TEVS = 1:1 after adsorption of P (**a**), 4-NP (**b**) and NB (**c**).

**Figure 16 molecules-26-02396-f016:**
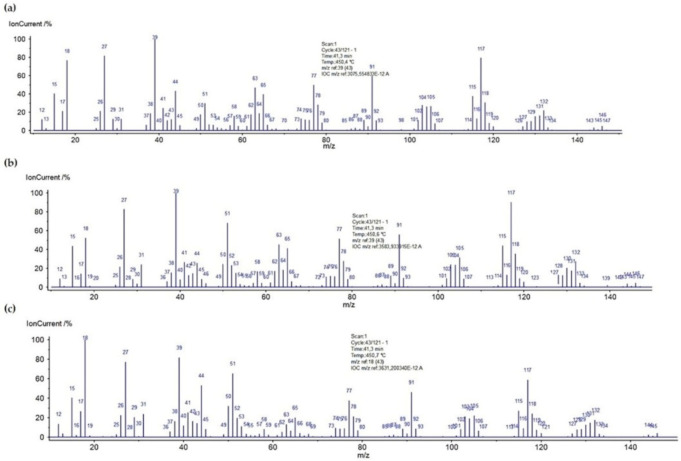
MS profiles registered during decomposition of DVB:TEVS = 1:1 after adsorption of P (**a**), 4-NP (**b**) and NB (**c**).

**Figure 17 molecules-26-02396-f017:**
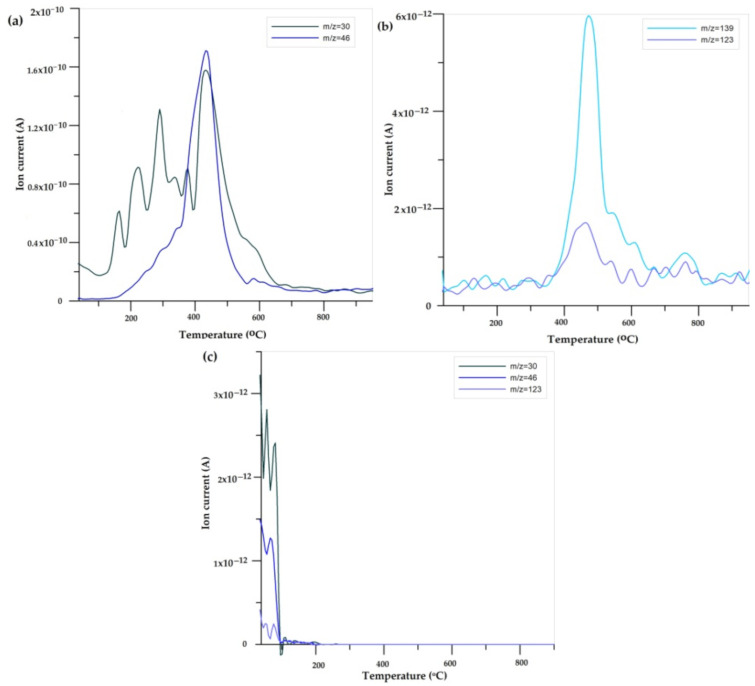
MS gaseous profiles registered during decomposition of DVB:TEVS = 1:1 after adsorption of 4-NP *m*/*z* = 30 and 46 (**a**) and *m*/*z* = 139 and 123 (**b**) and of NB *m*/*z* = 30, 46 and 123 (**c**).

**Figure 18 molecules-26-02396-f018:**
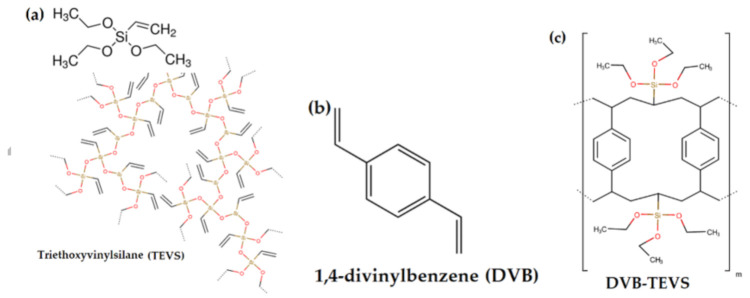
The chemical structure of inorganic (**a**) and organic (**b**) components of hybrid materials as well as the schematic structure of the copolymer (**c**).

**Table 1 molecules-26-02396-t001:** Structural parameters of the investigated organic-inorganic systems.

Sample	R ^a^[Å]	PDDF ^b^ [Å]	D_max_ (PDDF) ^c^ [Å]	R_g_[Å] (Sphere) ^d^		Porod Approximation				Specific Surface Area	
				PDDF ^e^	Guinier ^f^	K ^g^	Q ^h^ [Å^−1^]	C_0_ ^i^	S/V [Å^−1^]	S_SAXS_ ^j^ [m^2^/g]	S_BET_ [m^2^/g]
DVB:TEVS = 1:2	45	110(Spherical) 45(Rod-type cross-section)	399	125	115	0.798	34.7	12.1	0.061	610	521
DVB:TEVS = 1:1	35	140	420	112	101	0.649	27.8	11.8	0.048	480	402
DVB:TEVS = 2:1	30	179	504	110	95	0.511	29.6	13.3	0.039	390	316

^a^ The volume-weighted particle size distribution Dv(R) as the maximum value of the function. ^b^ Pair distance distribution function PDDF as the maximum value of the function. ^c^ Maximum dimension D_max_ means also R-value (distance) at which PDDF goes to 0. This parameter is defined as diameter across the longest dimension of the particles and is zero for r >D_max_. ^d^ Radius of gyration as the mean square distance from the center of their distribution. R_g_ provides a measure of the overall size of the scattering objects. ^e^ R_g_ determined from p(r) function is proportional to the normalized second moment of p(r) (Equation.(4)) from the whole scattering curve. ^f^ The Guinier plot, as ln(I(q)) vs. q^2^ was used to determine the R_g_ from the slope of the Guinier plot. ^g^ Porod constant is proportional to the surface area and the square of the electron density contrast. ^h^ Scattering invariant Q is proportional to the mean-square density fluctuation of scattering volume. Q = 2π^2^·Δρ^2^·V where volume V and scattering contrast Δρ. For calculation Q invariant the scattering intensities to q = 0 and also towards large q should be extrapolated. ^i^ Bacground constant which illustrates asymptotic decay of the SAXS curve at the high q values. ^j^ Surface area by SAXS calculated by Equation (2).

**Table 2 molecules-26-02396-t002:** The values of parameters characterizing the porous structure of DVB-TEVS.

Sample	Surface Area (S_BET_) [m^2^/g]		Pore Volume [cm^3^/g]		Pore Size [nm]	
	S_BETTotal_ ^a^	S_MIC_ ^b^	V_Total_ ^c^	V_MIC_ ^d^	D_h_ ^e^	BJH_ADS_ ^f^
DVB:TEVS = 1:2	521	-	0.84	-	6.4	4.8
DVB:TEVS = 1:1	402	2.5	0.74	-	7.4	5.1
DVB:TEVS = 2:1	316	27	0.54	0.01	6.8	5.8

^a^ S_BET_, the BET specific surface area; ^b^ S_MIC_, the micropore surface area; ^c^ V_t_, the total pore volume; ^d^ V_MIC_, the micropore volume; ^e^ D_h_, the average hydraulic pore diameter (4V/A); ^f^ BJH_ADS_, average BJH adsorption pore diameter.

**Table 3 molecules-26-02396-t003:** Relative standard deviations SD(c)/co (%) for m-exp, FOE, SOE, MOE, f-FOE, f-SOE, F-MOE, McKay pore diffusion (PDM) and IDM model (Crank).

System	m-exp (%)	FOE (%)	SOE (%)	MOE (%)	f-FOE (%)	f-SOE (%)	f-MOE (%)	IDM (%)	PDM (%)
NB/DVB:TEVS = 2:1	0.544	3.200	0.703	0.935	1.281	0.635	0.651	5.81	24.78
NB/DVB:TEVS = 1:2	0.469	2.480	0.695	0.699	1.310	0.498	0.501	7.21	25.92
NB/DVB:TEVS = 1:1	0.205	1.227	0.995	0.999	0.473	0.581	0.395	15.36	24.75
P/DVB:TEVS = 2:1	0.805	0.802	0.818	0.794	0.685	0.680	0.684	10.36	23.89
P/DVB:TEVS = 1:2	0.704	0.776	0.782	0.772	0.763	0.750	0.711	9.06	22.56
P/DVB:TEVS = 1:1	0.506	1.455	1.450	1.459	1.330	1.351	1.351	6.87	18.96
4-NP/DVB:TEVS = 2:1	0.397	1.277	0.791	0.796	0.407	0.427	0.408	8.78	29.56
4-NP/ DVB:TEVS = 1:2	0.572	0.801	0.704	0.708	0.750	0.704	0.709	9.64	28.25
4-NP/DVB:TEVS = 1:1	0.669	3.519	4.949	1.889	1.737	0.999	9.694	13.53	21.64

**Table 4 molecules-26-02396-t004:** Optimized parameters of m-exp equation.

System	*f*_1_, log *k*_1_	*f*_2_, log *k*_2_	*f*_3_, log *k*_3_	*u_eq_*	*t*_1/2_ (min)	*S*D(*c*)/*c*_o_ (%)	1-*R^2^*
NB/DVB:TEVS = 2:1	0.095,0.116	0.627,−1.90	0.278,−2.81	0.805	73.1	0.544	5.2·10^−4^
NB/DVB:TEVS = 1:2	0.080,−0.967	0.715,−1.968	0.205,−2.851	0.788	75.9	0.469	9.6·10^−4^
NB/DVB:TEVS = 1:1	0.056,0.187	0.686,−1.809	0.258,−2.451	0.553	55.9	0.205	4.2·10^−4^
P/DVB:TEVS = 2:1	0.005,−0.788	0.995,−4.881	-	0.148	52334.6	0.805	5.4·10^−2^
P/DVB:TEVS = 1:2	0.018,−0.889	0.982,−4.939	-	0.154	52334.6	0.704	5.1·10^−2^
P/DVB:TEVS = 1:1	0.056,−1.452	0.006,−0.816	0.938,−4.870	0.168	46568.7	0.506	1.6·10^−2^
4-NP/DVB:TEVS = 2:1	0.124,−1.332	0.368,−2.518	0.508,−3.380	0.256	483.9	0.397	2.3·10^−3^
4-NP/DVB:TEVS = 1:2	0.130,1.063	0.450,−2.839	0.370,−4.040	0.180	803.6	0.572	1.4·10^−2^
4-NP/DVB:TEVS = 1:1	0.049,−0.071	0.598,−1.607	0.353,−2.972	0.499	51.8	0.669	2.4·10^−3^

**Table 5 molecules-26-02396-t005:** TG, DTG and DSC data obtained in helium atmosphere for DVB-TEVS materials.

Sample	TG [%]				DTG	DSC	
	m_IDT_ (170–330 °C)	m_loss_ 330–550 °C	m_loss_ 550–950 °C	m_loss_ TOTAL	T_d_ [°C]	T_d_ [°C]	ΔH_d_ [J/g]
DVB:TEVS = 1:2	1.39	83.99	1.47	86.85	453	453	150.8
DVB:TEVS = 1:1	0.92	83.87	1.89	86.68	450	453	145.7
DVB:TEVS = 2:1	0.36	85.06	2.13	87.55	453	453	147.4

**Table 6 molecules-26-02396-t006:** TG, DTG and DSC data obtained in helium atmosphere for DVB-TEVS materials with loaded P, 4-NP and NB.

Sample	TG [%]				DTG	DSC	
	m_IDT_ (170–330 °C)	m_loss_ 330–550 °C	m_loss_ 550–950 °C	m_loss_ TOTAL	T_d_ [°C]	T_d_ [°C]	ΔH_d_ [J/g]
P_DVB:TEVS = 1:2	2.61	75.69	3.11	81.41	450	452	164.8
P_DVB:TEVS = 1:1	3.41	75.86	3.60	82.87	450	452	167.1
P_DVB:TEVS = 2:1	3.98	76.76	2.73	83.47	450	452	198.2
NP_DVB:TEVS = 1:2	3.70	72.62	8.50	84.82	440	449	191.1
NP_DVB:TEVS = 1:1	3.19	72.51	6.02	81.72	449	449	134.5
NP_DVB:TEVS = 2:1	3.25	72.53	5.96	81.74	449	449	135.8
NB_DVB:TEVS = 1:2	1.24	80.20	2.97	84.41	450	450	170.2
NB_DVB:TEVS = 1:1	0.67	83.25	2.67	86.59	449	450	199.5
NB_DVB:TEVS = 2:1	1.29	81.61	2.90	85.80	448	450	186.9

**Table 7 molecules-26-02396-t007:** The physicochemical properties of used adsorbates.

Adsorbate	Chemical Formula	Molecular Weight [g/mol]	Water Solubility [g/100 mL at 20 °C]	Ionization Constant pKa	Melting Point [°C]	Boiling Point [°C]	Chemical Safety
Phenol (P)	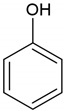	94.11 ^1^	8.3 ^1^	9.99 ^1^	40.5 ^1^	181.7 ^1^	Corrosive Acute toxic Health hazard ^1^
Nitrobenzene (NB)	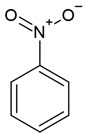	123.11 ^1^	0.19 ^1^	---	5.7 ^1^	210.9 ^1^	Acute toxic Health hazard ^1^
4-Nitrophenol (4-NP)	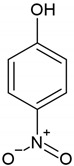	139.11 ^1^	1.6 ^1^	7.15 ^1^	113 ^1^	279 ^1^	Irritant Health hazard ^1^

^1^https://pubchem.ncbi.nlm.nih.gov (accessed on 01 February 2021).

**Table 8 molecules-26-02396-t008:** Compositions of reaction mixtures in organic-inorganic materials synthesis.

Sample	Monomers		Pore-Forming Diluents	
	DVB-TEVS Molar Ratios		Toluene and Decano-1-ol Volumes [cm^3^]	
DVB:TEVS = 1:2	1	2	5	5
DVB:TEVS = 1:1	1	1	5	5
DVB:TEVS = 2:1	1	0.5	5	5

## Data Availability

The data are available by corresponding author.

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
