# Peer review of "Physicochemical and Adsorption Characteristics of Divinylbenzene-co-Triethoxyvinylsilane Microspheres as Materials for the Removal of Organic Compounds"

_molecules, 2021, doi:10.3390/molecules26082396_

Round 1

Reviewer 1 Report

This manuscript deals with preparation and characterization of a new material for solid-phase extraction of various groups of pollutants in water remediation. In particular, porous microspheres of divinylbenzene-triethoxyvinylsiloxane (DVB/TEVS) copolymer have been synthesized by the suspension-polymerization technique at three different DVB/TEVS molar ratios (1:2, 1:1, and 2:1). This copolymer was obtained in form of micrometric spheres showing a well developed porous structure that has been characterized by scanning electron microscopy (SEM) and porosity measurements. In addition, the achieved copolymer was characterized by FT-IR/ATR spectroscopy, small angle X-ray diffraction analysis (SAXS), nitrogen adsorption/desorption, and thermal analysis (TG, DTG, DSC). Then, the adsorption properties toward nitrobenzene, nitrophenol, and phenol were investigated on the basis of kinetic measurements and the best adsorption performance was found for the nitrobenzene molecule, using the most hydrophobic DVB/TEVS combination.

This manuscript includes useful technical information about the development of an effective, inexpensive and selective polymeric adsorbent for wastewater remediation. The manuscript has been carefully prepared and is of good overall quality, however the Introduction Section must be organized in a better way and more appropriate references must be included in it. In fact, the obtained copolymer has a complex texture made of organic-inorganic microscopic phases (more properly, hydrophilic/hydrophobic micro-phases, since also organosilanes are organic compounds), however it cannot be described as ‘composite materials’ (that is, a filled polymer), like authors have extensively done in the Introduction Section. The description given in the Introduction Section is too close that valid only for traditional composite systems and not adequate for the prepared copolymer. This complex substance should be classified as hybrid material (see for example: Leno Mascia et al., Macromol. Symp. 2007, 247, 129-139) and enough references concerning the suspension-polymerization technique, some previous literature results about DVB-TEVS copolymers, and hybrid organic-inorganic materials should be included.

Author Response

The answers to Reviewer 1 comments are in the file R1_answer_molecules-1145552.pdf

Reviewer 2 Report

Lines 242-244 When the ratio of organic to inorganic phase equals1:2, the spherical morphology of the material is disturbed with many deviations from the ideal spherical form, the material is heterogeneous, several defects are observed(Figure 2a-c).The most regular shape, the highest homogeneity,and the lowest porosity wasob-245served for DVB:TEVS=2:1 sample (Figure2g-i). In this case,the surface of the spheres is the most uniform and smooth.The sample, DVB:TEVS=1:1,shows intermediate morpho-logical and pore properties in comparison the materials synthesized at proportionsDVB:TEVS=2:1 and DVB:TEVS=1:2(Figure 2d-f). The use of a silane coupling agent with a cross-linking nature is responsible for the construction of a porous network.

 Comment Only description of the observation

Line 129 Comment  Sample preparation for SEM and Nitrogen Adsorption-Desorption Measurements should be described in materials and Methods.

Lines 314- 318The lack of a sharp interference peak on the SAXS profile suggests the absence of the regular superstructural forms of domains. The experimental scattering curve includes the sum of the scattering of various phases and their interactions. The nature of the scattering curve is similar for all studied samples. However, some differences in the level of scattering were observed. It was found that the introduction of the greater amount of TEVS into the composite body raises the intensity of scattering at low-angles.

References must be included to support the statement.

Lines 325-328 increased porosity is associated with a higher amount of crosslinking agent and may cause the presence of additional nanometer forms (pores) able to generate the scattering effect. In this way, a comparison of the level of scattering by hybrid samples can provide information about the degree of surface homogeneity. Related references must be included to support the explanation of phenomenon.

Author Response

The answers to Reviewer 2 comments are in the file R2_answer_molecules-1145552.pdf

Round 2

Reviewer 1 Report

The manuscript has been extensively improved. In particular, the developped material has been correctly classified and the principal synthesis techniques for preparing these materials have been indicated in the Introduction Section of the revised manuscript version. In addition, an adequate number of references have been added. Therefore, the manuscript can be accepted for publication in its present form.

Reviewer 2 Report

 Line 103 poly(vinyl alcohol) (PVA) were purchased in Fluka AG Molecular weight and degree of deacetylation should be included